# The Control Strategy and Kinetics of VFAs Production in an ASBR Reactor Treating Low-Strength Mariculture Wastewater

**DOI:** 10.3390/ijerph19137858

**Published:** 2022-06-27

**Authors:** Fan Gao, Cuiya Zhang, Qinbang Sun, Guangjing Xu

**Affiliations:** 1National Marine Environmental Monitoring Center, Dalian 116023, China; gaofan@nmemc.org.cn (F.G.); qbsun@nmemc.org.cn (Q.S.); 2College of Marine Technology and Environment, Dalian Ocean University, Dalian 116023, China; 3State Environmental Protection Key Laboratory of Coastal Ecosystem, Dalian 116023, China; 4College of Ocean and Civil Engineering, Dalian Ocean University, Dalian 116023, China; zhangcuiya@dlou.edu.cn; 5Key Laboratory of Nearshore Marine Environmental Science and Technology in Liaoning Province, Dalian Ocean University, Dalian 116023, China

**Keywords:** anaerobic fermentation, decarbonization, low-strength mariculture wastewater, kinetics parameters

## Abstract

As an environment-friendly wastewater treatment process, the anaerobic fermentation process has been widely used for the pretreatment of high-strength wastewater. However, it is rarely applied to treat low-strength wastewater due to low methane recovery. In this study, anaerobic fermentation treating low-strength mariculture wastewater was studied in an anaerobic sequencing batch reactor (ASBR) with a COD removal rate of 75%. Anaerobic fermentation was successfully controlled at the acidification stage by increasing COD loading. As the greenhouse gas emission decreased, the residual organics were enough for biological nutrients’ removal. Fluorescence in situ hybridization results showed that the dominant bacteria in the ASBR were acidogenic bacteria and methanogens, accounting for 39.7% and 46.5% of the total bacteria, respectively. Through the calculation processing of the experimental data, the order of the anaerobic fermentation reaction was a second-order reaction. The kinetic parameters of low-strength organic wastewater treatment were determined by using the Grau second-order substrate removal model, Stover–Kincannon model, Monod model and Haldane model. The maximum rate removal constant U_max_, sludge yield coefficient Y and inhibition constant K_i_ were 1.157 g/(L·d), 0.153 mgVSS/mgCOD and 670 mg/L, respectively. It provided data support for the practical application of the anaerobic fermentation treating low-strength wastewater.

## 1. Introduction

The mariculture process is a production mode of breeding marine economic animals and plants using shallow sea, tidal flat, pond and other sea areas, which is the main way of human directional utilization of marine biological resources [1,2]. Mariculture plays an important role in promoting the economic development of coastal areas and enriching the human food structure, but the seawater pollution caused by mariculture wastewater has attracted more and more attention [3]. Mariculture wastewater is rich in solids, organic matters, nitrogen and phosphorus pollutants, which mainly originate from the large amount of unused bait and excrement of mariculture organisms [4]. As the untreated mariculture wastewater is directly discharged into the sea, the surrounding seawater quality and marine ecological balance would be affected [5]. Some physical treatment processes including filtration, sedimentation and membrane separation are often used to remove suspended solids from mariculture wastewater [6,7]. The biological treatment methods were always used to simultaneously remove organic matters, nitrogen and phosphorus compounds after physical treatment processes [8].

Compared with municipal wastewaters and industrial wastewaters, high-salinity mariculture wastewaters contain small amounts of organic matter, nitrogen and phosphorus from residual feeds and feed feces [4]. The organic matters in mariculture wastewater are usually absorbed and utilized by microorganisms or directly converted into carbon dioxide and discharged into the atmosphere [9]. However, the random discharge of carbon dioxide is inconsistent with the requirements of carbon emission reduction and carbon neutralization in China [10]. Normally, organic carbons are used for the denitrification of nitrate produced by nitrification using ammonium in wastewater. The nitrogen removal efficiency is highly affected by the amounts of organic carbon sources in the mariculture wastewater treatment system (MWSY) [11]. It is necessary to maintain a certain concentration of chemical oxygen demand (COD) to nitrogen ratio (COD/N) for nitrogen pollutants’ removal [11,12]. Meanwhile, denitrification rates are affected by the type of carbon source [13]; that which had high biodegradability could be used for enhancing nitrogen removal efficiency in the MWSY. It was demonstrated that volatile fatty acids (VFAs) could not only clearly reveal the link between pH and bacterial composition [14], but also are the most favorite organic carbon sources for denitrifying bacteria [15].

Anaerobic fermentation processes are often used as a pretreatment process of industrial wastewater with a high organic load to enhance the biodegradability of wastewater. However, it was scarcely studied as a pretreatment process to treat mariculture wastewater with a low organic load. In addition, the results of the stoichiometry calculation showed that carbon removal via hydrolytic acidification could efficiently reduce 30–50% of greenhouse gas (CO_2_ and methane) emission, which was beneficial to carbon emission reduction; however, it was difficult to accurately control the carbon degradation of low-strength mariculture wastewater at the stage of acidification, otherwise the produced VFAs were transferred to biogas [16]. Among the anaerobic processes, the anaerobic sequencing batch reactor (ASBR) offers advantages of continuous treatment, great process flexibility and lesser space needs [17] and is suitable for the treatment of mariculture wastewater.

In this study, the anaerobic fermentation process was studied to treat low-strength mariculture wastewater in an ASBR reactor to explore the improvement direction of the mariculture wastewater treatment process under the background of carbon emission reduction. This study focused on how to control the VFAs’ production process and reduce greenhouse gas emission during the long-term operation of ASBR, and it determined the kinetics of the anaerobic fermentation reaction to further discuss the feasibility of such a process as a pretreatment process treating mariculture wastewater. The microbial community was also analyzed by the fluorescence in situ hybridization (FISH) technology.

## 2. Materials and Methods

### 2.1. Configuration and Operation of ASBR

An anaerobic fermentation process was performed in an ASBR treating low-strength mariculture wastewater (Figure 1). The ASBR was made of plexiglass with a diameter of 125 mm, an effective height of 270 mm and a working volume of 13.2 L. The experimental device also included a temperature controller, water outlet solenoid valve, time relay, electromagnetic stirrer, liquid level meter, peristaltic pump, water inlet bucket and water outlet bucket. The time relay control realized the alternating change of the ASBR working cycle. The microorganism was suspended in the reactor by electromagnetic stirring, and the stirring speed was 35 RPM. The temperature was controlled at about 30 °C. The peristaltic pump and solenoid valve controlled the inlet and outlet of the reactor, respectively.

The ASBR was operated under a different hydraulic retention time (HRT: 0.852 d, 0.697 d, 0.465 d, 0.387 d), and each HRT operation lasted for 9 days. The one-cycle timetable was shown in Table 1. Taking an example of HRT = 0.465 d, one ASBR cycle sequentially included 0.5 h of feeding, 9 h of fermentation, 2 h of sedimentation and 0.5 h of discharge, and meanwhile only fermentation duration was changeable for HRT optimization. The organic load of ASBR increased from 0.176 gCOD/(L·d) to 0.602 gCOD/(L·d).

### 2.2. Seeding Sludge and Synthetic Wastewater

The seeding sludge of ASBR was taken from the Lingshui Bay of Dalian. According to the actual composition and content of the main pollutants in mariculture wastewater, the specific formula was: 177–314.5 mg/L sucrose, 38.2 mg/L NH_4_Cl, 14.7 mg/L K_2_HPO_4_·3H_2_O, 220 mg/L NaHCO_3_, 101.5 mg/L MgCl_2_·6H_2_O, 55.5 mg/L CaCl_2_, 3.555 mg/L FeCl_2_·4H_2_O, 25–30‰ salinity and 1 mL/L of trace element solution, which contained 0.4 g/L CoCl_2_·6H_2_O, 0.81 g/L NiCl_2_·6H_2_O, 0.25 g/L Na_2_MoO_4_·2H_2_O, 0.21 g/L ZnCl_2_ and0.36 g/L MnCl_2_·4H_2_O.

### 2.3. Analytical Methods

The influent and effluent samples were collected on a daily basis. The samples were immediately analyzed or stored in a refrigerator at 4 °C until the analysis was carried out. COD, pH, SS and VSS were analyzed according to the standard methods for the examination of water and wastewater (20th ed.) [18]. The gas samples were collected on a daily basis by the gas sampling port of ASBR. VFAs, methane and carbon dioxide were measured by gas chromatography (GC-14C, SHIMADZU, Tokyo, Japan) with a thermal conductivity detector and a double injector connected to three 5 m length, 3 mm diameter columns with helium as carrier gas. The first column was a 5A 100:120 chromatographic column for giving CH_4_ concentrations; the second column was TDX-01 to give CO_2_ concentration; the third column was GDX-102 chromatographic column used to give the VFAs concentration.

### 2.4. SEM Analysis and FISH Analysis

The surface morphology and structure of anaerobic sludge collected in the 36th day were observed by scanning electron microscopy (SEM, FEI QUANTA 450, Thermo Fisher Scientific, Waltham, MA, USA). The SEM analysis experiment included fixation, cleaning, dehydration, drying, metal spraying and observation. The minimum separation rate of scanning electron microscopy was 6 nm.

On the 36th day, the sludge was also analyzed by FISH with eubacterial probe EUB338, acidogenic probe BAC307 and methanogenic probe MS1414 to determine the species and quantity of flora [19,20,21]. The color of the EUB338 probe was green, which represented the universal bacteria. The color of the BAC307 probe was purple, which represented acid producing bacteria. The color of the MS1414 probe was red, which represented methanogens. The FISH analysis experiment included fixation, cleaning, in situ hybridization, probe cleaning, observation and data analysis. In this study, the used 16S rRNA-targeted oligonucleotide probes were supplied by Takara (Dalian) Co., Ltd. (Dalian, China). The samples were observed by a fluorescence microscope (BX51, Olympus, Japan) in this study. The hybridization conditions of the above probes are as shown in Table 2.

### 2.5. Model Description

In this study, four mathematical models were applied to decide the kinetic parameters of ASBR, namely two linear models (Grau second-order substrate removal model and Stover–Kincannon model) and two nonlinear models (Monod model and Haldane model) (Table 3).

## 3. Results and Discussion

### 3.1. The Performance of ASBR

According to the change of COD loading, the experimental process was divided into five stages (Figure 2). It took about 15 days for the start-up of the ASBR. In this section, the decarbonization process of ASBR was studied, which included the acidogenesis and methanogenesis process. Additionally, the decarbonization efficiency was defined as the COD removal efficiency from wastewater. In the initial stage (day 0–15, stage I), the HRT was 0.852 d and the average influent COD was 150.2 mg/L. On the 6th day, the COD removal rate of the reactor decreased, and the VFA content of the effluent COD also decreased (Figure 2A). Meanwhile the pH value showed a downward trend from 7.8 to 7.16, and the sludge color gradually changed from brown to black. This might be caused by the death of aerobic bacteria in the inoculated sludge under anaerobic conditions, which brought a certain load impact to the ASBR, resulting in the increase in effluent COD concentration. As shown in Figure 2B, the pH value always showed a downward trend in the first stage, which indicated that the acid production process of the anaerobic fermentation reaction had been in progress and a certain amount of VFA accumulation was generated. From the 9th day on, the production of methane gas showed an upward trend. As the anaerobic fermentation process provided enough metabolic substrate VFA for the methanogenic process, the methanogenic phenomenon began to appear in the ASBR with the accumulation of VFA. On the 15th day, the effluent COD of the ASBR was only 27.7 mg/L with a COD removal efficiency of 76.6%, indicating that the fermentation was successfully started up.

In stage II (18–33 d), the average influent COD concentration increased to 175 mg/L without changing the HRT. As shown in Figure 2, during the 18th–24th days, the VFA concentrations gradually increased, and meanwhile the pH value gradually decreased, indicating that the acidogenesis process was the dominant reaction at this time. In the 24th–33rd days, the pH value showed an upward trend for the first time, while the VFA contents in the effluent COD began to decrease. Meanwhile, the COD removal efficiency and methane production rate of the ASBR gradually increased. It demonstrated that the methanogenic reaction was gradually strengthened with an increment of organic loading. The reason should be that the generation cycle of acidogenic bacteria was less than that of methanogenic bacteria. Long HRT and overloading resulted in the proliferation of methanogens in the ASBR. Therefore, the influent COD of 175 mg/L and HRT of 0.852 d were not conducive to VFA accumulation.

In stage III (36–60 d), the decarbonization performance and VFA accumulation of ASBR were investigated by simultaneously increasing COD concentrations to 198 mg/L and shortening HRT to 0.697 d (36–42 d), 0.465 d (45–51 d) and 0.387 d (54–60 d), respectively. As shown in Figure 2, on the 36th day, both the VFA production rate and the VFA percentage content in the effluent COD increased, while all the COD removal rates, pH values and methane production of the ASBR decreased. It indicated that VFA should accumulate with the COD loading increasing and HRT shortening. This was because the growth rate of acidogenic bacteria was higher than methanogens, and the substrate in the ASBR was always sufficient for the acidification with the increase in COD loading [22]. Furthermore, the increased VFA accumulation caused a pH decrease, which had an obvious inhibitory effect on the activity of methanogenic bacteria [23]. Therefore, the increase in the COD concentration and the shortening of HRT had no impact on the acetate production process, while it had a great impact on the methanogenic process. Moreover, the COD removal efficiency, pH value and VFA content in the COD effluent changed a little in 45–60 days, but the methane production obviously decreased. It indicated that shortening HRT had no impact on the accumulation of VFA in the ASBR.

In stage IV (63–78 d), the influent COD concentration increased to 220 mg/L without an HRT change. In days 63–69, the effluent concentrations of VFA increased with a COD removal rate decreasing by 10%, and the production of methane was nearly unchanged. This showed that the increase in influent COD concentration resulted in high VFA accumulation. However, due to the short HRT (0.387 d), the methanogenic process was inhibited, resulting in the decline of the decarbonization efficiency of ASBR. In days 72–78, with HRT increasing to 0.697 days, the COD removal rate of ASBR increased but the effluent VFA concentration decreased. It indicated that the long HRT was helpful for the methanogenic process and could improve the decarbonization performance of ASBR. From all the results above, it could be seen that the influent COD concentration and HRT played key roles in VFA accumulation.

In stage V (81–90 d), the maximum decarbonization performance of ASBR was studied by increasing the influent COD concentration. In this stage, the influent COD concentration of the reactor increased from 232 mg/L to 266.4 mg/L, while the HRT was kept as 0.852 d. The ASBR showed a very good decarbonization performance, and the COD removal efficiencies were kept above 80%. As autotrophic and heterotrophic methanogenic bacteria use H_2_/CO_2_ and volatile fatty acids as energy materials, respectively, both volatile fatty acids and carbon dioxide produced in the acid production process could be used by methanogenic bacteria. With the influent COD increasing, the volume ratio of methane to carbon dioxide (CH_4_/CO_2_) increased, while the effluent concentration of VFA slightly decreased. The experimental results showed that the single change of COD concentration in ASBR was not conducive to the accumulation of VFA. The volume ratio of methane to carbon dioxide was 0.15~0.2 in stages III and IV, which reduced 50% of methane emission compared to stage V. Both methane and carbon dioxide are the main greenhouse gases of the Earth’s atmosphere, and the greenhouse effect of methane is 22 times higher than that of carbon dioxide [24]. Through controlling VFA production, the reduction of the emission of methane and the greenhouse effect was realized in this study.

### 3.2. Cyclic Concentration Profiles of COD and VFA in One ASBR Cycle

The concentration profiles of COD and VFA in one ASBR cycle (HRT = 0.465 d) are shown in Figure 3. According to the fitting results of the experimental data, the profiles of COD concentration were approximate to the negative first-order exponential function. In addition, through the derivation and deformation of the fitted relationship formula between COD concentration and time in the ASBR (Figure 3), the relationship formula of COD removal efficiency and concentration in the ASBR was obtained as follows:(1)$$-\frac{dC}{dt}=1.62\times 10^{-3}C^{2.2}$$

In Formula (1), *dC/dt* is the COD removal rate; t is the time; C is the COD concentration in ASBR. According to Formula (1), the COD removal reaction order of the reactor was determined as an approximate second-order, which was coincident with the characteristics of the anaerobic biological reaction that included fermentation reaction (acid production process) and methane production reaction [22]. According to Figure 3, the COD concentration of the ASBR decreased rapidly from 198.6 mg/L to 70 mg/L in the first 3 h, while the COD concentration had no obvious change after 8 h. In the first 3 h, the VFA concentration of the ASBR had a wavy change, and the maximum VFA accumulation concentration arrived at 113.6 mg/L. The anaerobic fermentation reaction was the dominant reaction in the ASBR at this time. It was demonstrated that the accumulation of VFA in the anaerobic fermentation reaction could be realized by controlling the reaction time as the first 3.6 h in one cycle.

The composition and concentration changes of VFA in the reactor (HRT = 0.465 d) in one cycle are shown in Figure 4. The results showed that the VFA of the ASBR was composed of acetate, propionate and butyrate. No formic acid was detected in the experiment. Acetate and propionate were the main VFA components in the reactor, accounting for 80% and 15% of the VFA, respectively. It was reported that anammox bacteria had an affinity for acetate and propionate [25,26]. Therefore, the anaerobic fermentation reaction could be used as the decarbonization pretreatment process of the anammox reaction. From Figure 4, only a small amount of acetate existed in the initial stage of the experiment. In the 2nd hour, the concentrations of acetate, propionate and butyrate increased, reaching 87.5 mg/L, 19.0 mg/L and 7.44 mg/L, respectively. It was demonstrated that a certain amount of VFA accumulated at this time. However, the acetate in the reactor decreased rapidly by 50% in the 3rd hour, which was due to the gradual enhancement of the methanogenic reaction with the accumulation of acetate, resulting in the rapid consumption of acetate. In addition, the concentrations of propionate and butyrate gradually decreased in the reactor with the passage of time, and butyrate was not detected after the 5th hour. This was mainly due to the fact that the fermentation reaction further changed propionate and butyrate into a small molecule acetate, which was gradually consumed as the substrate of methanogenesis. In one cycle of the experiment, acetate was the main component of VFA, and its change trend was similar to that of the VFA in Figure 3.

### 3.3. SEM and FISH Analysis

At the beginning of the inoculation, the sludge was dark brown and flocculent (as shown in Figure 5A) with a musty smell. The reactor gradually turned black and gray on the 6th day, and the sludge of the reactor completely turned black on the 15th day and the sludge gradually gathered to form small particles. On the 36th day, the sludge in the reactor was analyzed by SEM. The results (as shown in Figure 5B) showed that the sludge in the reactor was mainly composed of cocci, bacilli and some filamentous bacteria.

The FISH results showed that there were methanogens (Figure 5D) and acidogenic bacteria (Figure 5E) in the reactor. The FISH images were quantitatively analyzed by Image Pro Plus software. The results showed that methanogens accounted for 46.5% of the whole bacteria and acidogenic bacteria accounted for 39.7% of the whole bacteria. The proportion of methanogens and acidogenic bacteria in the whole bacteria was not much different. On the 36th day, the number of methanogens was slightly higher than that of the acidogenic bacteria. The acetogenic bacteria could further change macro-molecular VFAs (propionate and butyrate) to acetate under anaerobic condition. Acetate is a favorite substrate of methanogens [27,28]. The number of methanogens increased with the accumulation of acetate. Therefore, VFAs’ concentration is a key impact factor for the abundance of methanogens.

### 3.4. Kinetics of Anaerobic Fermentation Reaction Treating Low-Strength Mariculture Wastewater

In order to accurately analyze the anaerobic fermentation process treating low-strength mariculture wastewater, the experimental data were simulated to determine the kinetic parameters using the four classical pollutant removal models of the Grau second-order model, Stover–Kincannon model, Monod model and Haldane model (Table 4). The results showed that the anaerobic fermentation reaction was highly consistent with the Grau second-order model and Stover–Kincannon model, with a simulation correlation coefficient R^2^ of 0.983 (Grau second-order model) and 0.985 (Stover–Kincannon model). The Grau substrate removal rate constant Ks_2_ was calculated as 0.288 d^−1^ close to the reported value of 0.337 d^−1^ in the literature [29]. The maximum rate removal constant U_max_ and the saturation constant K_B_ were 1.157 g/(L·d) and 1.132 g/(L·d), respectively, which were lower than those under the high organic carbon load [30].

The sludge yield coefficient Y and the endogenous decay coefficient K_d_ were calculated as 0.153 mgVSS/mgCOD and 3.67 × 10^−3^ d^−1^ through the Monod model. The Y value of this study was higher than the reported one of methanogens (0.01–0.05 mgVSS/mgCOD), while the one was the same as that of acidogenic bacteria (0.14–0.17 mgVSS/mgCOD) [31]. The two results above showed that acidogenic bacteria and methanogens existed in the reactor, but the abundance of acidogenic bacteria should be higher than that of methanogens. Moreover, the maximum specific growth rate μ_m_ was calculated as 0.053 d^−1^ through the Monod model, which was similar to the reported prescribed minimum of methanogens (0.041–0.912 d^−1^) [32,33]. This further verified the existence of other flora in the reactor. Moreover, the inhibition constant K_i_ was calculated as 670 mg/L. The kinetic parameters provided an important basis for the stable operation of the ASBR treating low-strength mariculture wastewater.

## 4. Conclusions

This paper studied the decarbonization performance of the anaerobic fermentation reaction and the change of VFA concentration in the ASBR treating low-strength mariculture wastewater and explored the method of effectively controlling greenhouse gas emissions in this process. The kinetics parameters were obtained by four different kinetic equations to supply the data basis for the practical application of the anaerobic fermentation process treating low-strength wastewater. The main conclusions were as follows:(1)The ASBR was successfully started under a low organic load, and the decarbonization effect of the reactor reached more than 75%. The influent COD concentration and HRT were the key factors affecting the VFA accumulation in the ASBR. The VFA accumulation could be realized by increasing the influent COD concentration and shortening the hydraulic retention time.(2)The reaction order of the anaerobic biological reaction was approximately 2 in this experiment. The VFA accumulation mainly occurred in the first 3 h, and the VFA accumulation reached the maximum in the 2nd hour. The main components of VFA in the reactor were acetate, propionate and butyrate. In the later stage of the reaction, only acetate and propionate existed in the reactor, and their contents accounted for more than 95% of the VFA. The FISH results showed that acidogenic bacteria and methanogens coexisted in the ASBR. The proportions of acidogenic bacteria and methanogens were 39.7% and 46.5%, respectively.(3)The experimental data of the reactor were simulated using the Grau second-order substrate removal model, Stover–Kincannon model, Monod model and Haldane model, and the kinetic parameters of the anaerobic fermentation reaction treating low organic load wastewater were obtained as the Grau substrate removal rate constant K_S2_ = 0.288 d^−1^, maximum rate removal constant U_max_ = 1.157 g/(L·d), saturation constant K_B_ = 1.132 g/(L·d), endogenous decay coefficient K_d_ = 3.67 × 10^−3^ d^−1^, sludge yield coefficient Y = 0.153 mgVSS/mgCOD and inhibition constant K_i_ = 670 mg/L.

## Figures and Tables

**Figure 1 ijerph-19-07858-f001:**
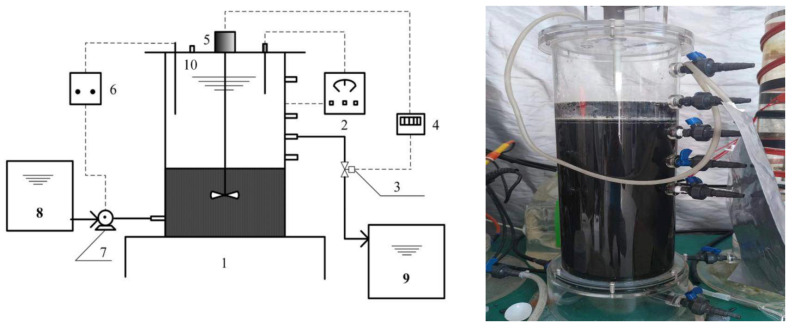
ASBR experimental device: 1: ASBR; 2: temperature controller; 3: solenoid valve; 4: time relay; 5: stirrer; 6: water-level meter; 7: peristaltic pump; 8: feeding tank; 9: effluent tank; 10: gas-sampling port.

**Figure 2 ijerph-19-07858-f002:**
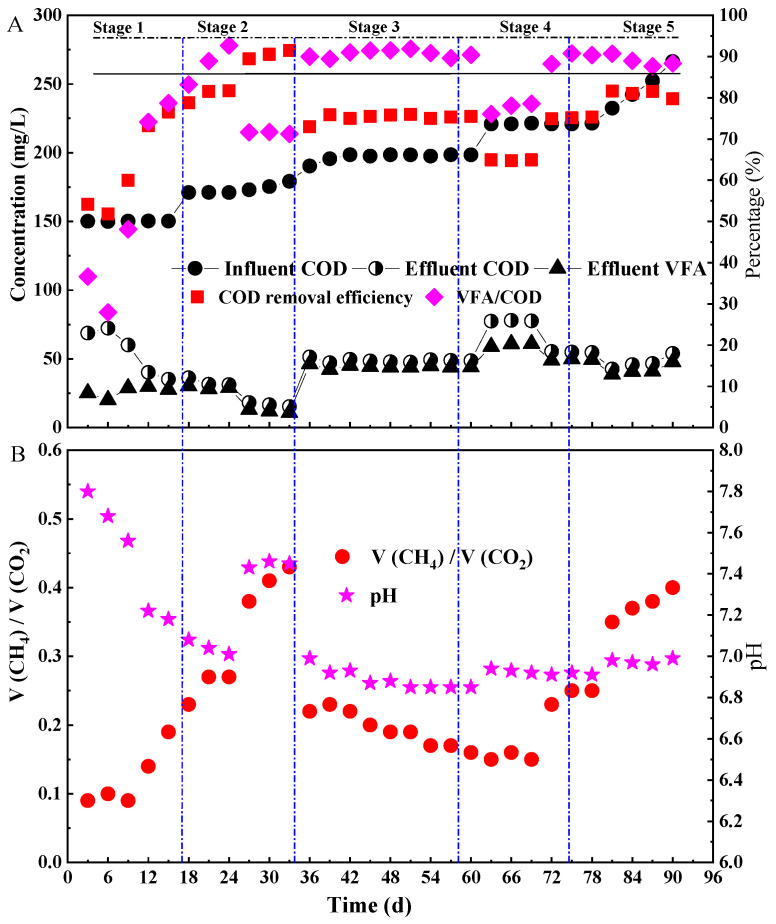
Performance of the ASBR. (**A**) COD removal and VFA production; (**B**) V(CH_4_)/V(CO_2_) and pH.

**Figure 3 ijerph-19-07858-f003:**
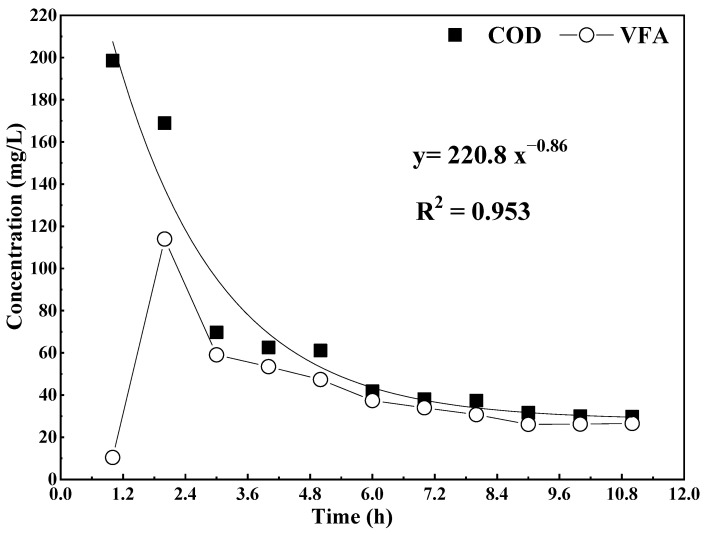
The concentration profiles of COD and VFAs in one ASBR cycle.

**Figure 4 ijerph-19-07858-f004:**
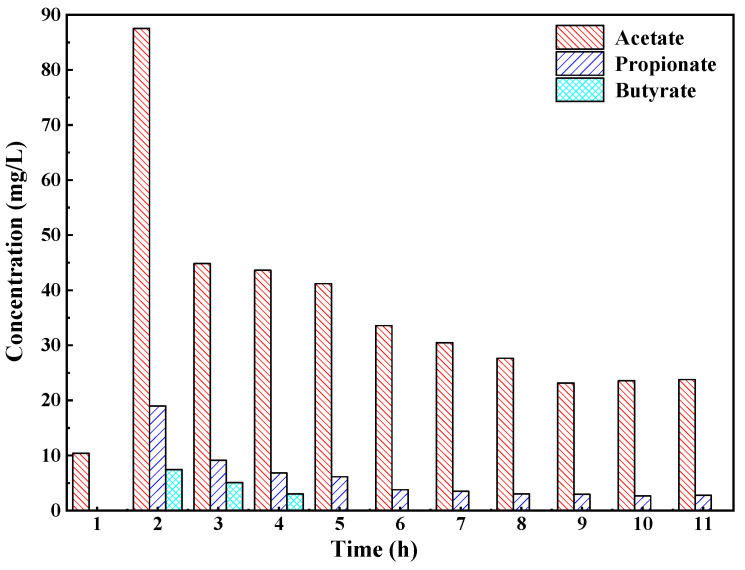
Cyclic concentration profiles of VFAs in one ASBR cycle.

**Figure 5 ijerph-19-07858-f005:**
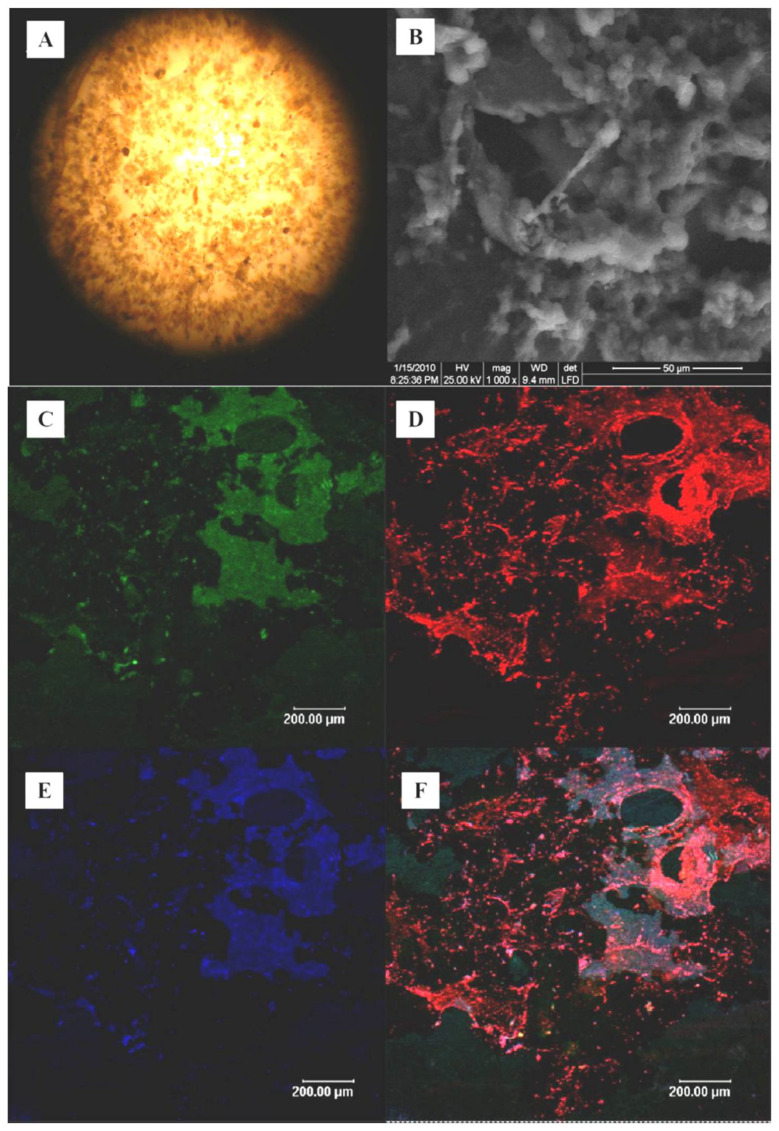
The photos of microbial flora analysis: (**A**) the microscope photos of seeding sludge; (**B**) the SEM images of bacteria on the 36th day; (**C**,**D**) the bacteria FISH analysis of ASBR on the 36th day: (**C**) Eubacteria; (**D**) Methanogens; (**E**) Acid-forming bacteria; (**F**) Superposition of three FISH images.

**Table 1 ijerph-19-07858-t001:** The one-cycle timetable for the ASBR operation under different HRT.

HRT (d)	Feeding Time (h)	Fermentation Time (h)	Sedimentation Time (h)	Discharge Time (h)
0.387	0.5	7	2	0.5
0.465	0.5	9	2	0.5
0.697	0.5	13	2	0.5
0.852	0.5	18	2	0.5

**Table 2 ijerph-19-07858-t002:** The 16S rRNA-targeted oligonucleotide probes and hybridization conditions.

Probe Name	Formamide(%)	Fluorescent Dye	Color	Excitation and Emission Wavelength (nm)
EUB338	20	FITC	Green	488, 528
MS1414	20	Cy3	Red	543, 570
BAC307	40	Cy3	Purple	630, 670

**Table 3 ijerph-19-07858-t003:** The kinetic models used in this study.

Model Name	Mathematical Equations	Parameters Meaning
Grau-2 substrate removal	$-\frac{dS}{dt}=K_{s2}\cdot X\cdot {\left(\frac{Se}{Si}\right)}^{2}$	*S_i_*, *S_e_*: Influent and effluent substrate concentration (mg/L), respectively *X*: Biomass concentration in a reactor (mg/L) *K_s2_*: Grau substrate removal rate constant (d^−1^)
Stover–Kincannon	${\left(\frac{dS}{dt}\right)}^{-1}=\frac{V}{Q\left(S_{i}-S_{e}\right)}=\frac{K_{B}}{U_{max}}\cdot \frac{V}{QS_{i}}+\frac{1}{U_{max}}$	*U_max_*: The maximum utilization rate constant g/(L·d) *K_B_*: The saturation value constant g/(L·d) *Q*: Flow rate of influent wastewater (L/d)
Monod	$\frac{dX}{dt}=\frac{Q}{V_{b}}\cdot X_{0}-\frac{Q}{V_{b}}\cdot X_{e}+\mu \cdot X-K_{d}\cdot X$ $\frac{dS}{dt}=\frac{Q}{V_{b}}\cdot S_{i}-\frac{Q}{V_{b}}\cdot S_{e}-\frac{\mu \cdot X}{Y}$ $\mu =\frac{\mu_{m}\cdot S_{e}}{K_{s}+S_{e}}$	*X_0_, X_e_*: Biomass concentration of influent and effluent wastewater (mg/L), respectively *V_b_*: Volume of sludge bed (L) *μ*: Specific growth rate (d^−1^) *K_d_*: Endogenous decay coefficient (d^−1^) *K_s_*: Half-velocity saturation constant (mg/L) *Y*: Cell yield coefficient (mg VSS/mg COD) *μ_m_*: Maximum specific growth rate (d^−1^)
Haldane	$\frac{V_{b}\cdot X}{Q\cdot \left(S_{i}-S_{e}\right)}=\frac{{\left(S_{e}\right)}^{2}}{K_{i}\cdot k}+\frac{K_{s}}{k}+\frac{S_{e}}{k}$ $k=\mu_{m}/Y$	*K_i_*: Inhibition constant (mg/L)

**Table 4 ijerph-19-07858-t004:** The calculated kinetic parameters of anaerobic fermentation reaction in this study.

Items	Kinetic Parameters and Units	Value	Fitting Results	Correlation Coefficient R^2^	Model Name
Grau substrate removal rate constant	Ks_2_, d^−1^	0.288	Y = 1.124X + 0.094	0.983	Grau-2
Maximum utilization rate constant	U_max_, g/(L·d)	1.157	Y = 0.978X + 8.64 × 10^−4^	0.985	Stover–Kincannon
Saturation value constant	K_B_, g/(L·d)	1.132			Stover–Kincannon
Endogenous decay coefficient	K_d_, d^−1^	3.67 × 10^−3^	Y = 6.438X + 0.024	0.835	Monod
Cell yield coefficient	Y, mgVSS/mgCOD	0.153			Monod
Half-velocity saturation constant	Ks, mg/L	79.88			Monod
Maximum specific growth rate	μ_m_, d^−1^	5.30 × 10^−2^			Monod
Inhibition constant	K_i_, mg/L	670	Y = 0.007X^2^ + 4.69X + 134.7	0.852	Haldane

## Data Availability

Not applicable.

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
