# Peer review of "The Control Strategy and Kinetics of VFAs Production in an ASBR Reactor Treating Low-Strength Mariculture Wastewater"

_ijerph, 2022, doi:10.3390/ijerph19137858_

Round 1
Reviewer 1 Report
General comments:
· There are no page numbers or line numbers in the manuscript, which made the review process difficult.
· Although the study reported in this manuscript contains some innovative aspects, the inadequacy of results and the poor discussion of reported results do not merit its publication in the present state. Major revisions including additional results and more in-depth discussion of results are needed before the manuscript can be considered for publication.
Specific Comments:
11. Section 2.4, paragraph 2: Not all bacteria are green. The authors should define the reference to “universal bacteria.”
22. Section 2.4, paragraph 2: References should be provided throughout this section whenever the authors refer to the color of bacteria.
33. Section 3: Please define the decarbonization efficiency which is frequently used in this section. In particular, the relationship between decarbonization and methanogenesis should be presented.
44. Section 3.1, lines 5-7: The authors should provide evidence for the stated hypothesis about the lysis of aerobic bacteria. How much did the COD concentration increase due to the lysis of cells? The increase of COD load due to the lysis of bacterial cells should not necessarily reduce the COD removal rate. There may be many other reasons for the decrease in the COD removal rate, such as low pH and accumulation of VFAs during the startup period. The hypothesis must be supported by experimental results and this phenomenon should be thoroughly discussed.
55. Page 5/13, paragraph 2: The authors state that anaerobic fermentation reaction was "violent”. This statement is confusing. The authors should clarify the sentence.
66. Page 5/13, paragraph 3: The authors should provide quantitative values for the generation cycle of acidogenic bacteria and methanogens either from their experimental results or from the literature.
77. Instead of “generation cycle” the authors should refer to the growth rate of bacteria or half life.
88. Section 3.2: It is stated that the profiles of COD concentration was fitted by first-order relationship (dC/dt = k C). Then, the authors state that a second-order relationship (Equation 3.1) was used for the COD removal efficiency and concentration in the ASBR. Please clarify.
99. Section 3.2, paragraph 2, lines 3-6: Please provide references for the order of reaction in anaerobic biological reaction.
110. Page 6/13, last line: It is stated that ”the accumulation of VFA in anaerobic fermentation reaction can be realized by controlling the reaction time”. The authors should elaborate on this statement. How can the reaction time be controlled? Are the authors referring to the control of operating conditions that affect the rate of reactions and thus the reaction time? This needs discussion and clarification.
111. Are the results presented in Figures 3 and 4 representative of the results obtained during the operation of ASBR at all different stages that used different operating conditions? Actually, this would be doubtful. The authors should present similar figures from different stages of ASBR operation to verify the pattern of change in the COD and VFA concentrations under all operating conditions (different stages of operation). The comparison of results in the suggested figures would provide very useful insight into the operation of ASBR and the activity of different bacterial cultures that control the acidogenic fermentation and methanogenic processes.
112. Section 3.3., paragraph 2: How about the presence of acetogenic bacteria that convert VFAs (propionate and butyrate) to acetate? The conversion of propionate and butyrate into acetate was discussed in the previous section.
113. Page 10/13: The authors state that the abundance of acidogenic bacteria should be higher than that of methanogens in their experimental system. Did the higher abundance of acidogenic bacteria occur under all experimental conditions? Is it desirable to have a higher concentration of acidogens compared to the methanogens? The authors should discuss these issues in the manuscript.
114. Section 4, Conclusions: The authors state that this study explored the method of effectively controlling greenhouse gas emissions. The emission of greenhouse gases should be analyzed and discussed in the manuscript with respect to the Equivalent-CO2 emission and the global warming potential of gases (CO2 and methane). This analysis was not conducted or reported in the manuscript.
Reviewer 2 Report
The following comments are helpful to enhance the quality of the paper.
- Do not use ‘study on’ in the title.
- What is the importance of ‘treating low-strength mariculture wastewater’? I think ‘high-strength’ is more important to treat. Why authors choose ‘low-strength’?
- Is the salinity effects on VFA production or pollutant removal?
Introduction
- Many biological treatments are mainly focused on organic matter removal. Only a few cases can remove nitrogen (anammox) and phosphorus (microalgae, cyanobacteria). Revise it.
- Is it any difference in mariculture type?
- Add the background about ASBR.
Material and methods
- Add the real picture in Fig. 1.
- Add the timetable for the ASBR operation.
- Add the exact location of ‘local offshore beach of Dalian’.
- Why do authors use ‘synthetic wastewater’? If wastewater was artificially mixed, it is not ‘low-strength mariculture wastewater’. It is just ‘artificial wastewater’.
- Add the number of standard methods in section 2.3.
- Add the analytical condition of GC. (Detector, column, and temperature).
- In section 2.4., why do authors use the word ‘activated sludge’? I think ‘anaerobic sludge’ is better.
- The quality of Table 2, 3 should be enhanced.
Results and discussion
- Why authors choose ‘acetate’ ‘propionate’ and ‘butyrate’ as VFAs?
Conclusion
- The sentences should be simplified.
Round 2
Reviewer 1 Report
The manuscript has been improved after the revision. The authors have incorporated my comments. The manuscript can be accepted in present form for publication.
Reviewer 2 Report
authors well revised the manuscript